# The impact of trait number and correlation on functional diversity metrics in real-world ecosystems

Timothy Ohlert[1]*, Kaitlin Kimmel[2], Meghan Avolio[3], Cynthia Chang[4], Elisabeth Forrestel[5], Benjamin P. Gerstner[6], Sarah E. Hobbie[7], Peter Reich[8,9,10], Kenneth D. Whitney[6], Kimberly Komatsu[11]

1 Department of Biology, Colorado State University, Fort Collins, CO, United States of America, 2 Global Water Security Center, University of Alabama, Tuscaloosa, AL, United States of America, 3 Department of Earth & Planetary Sciences, Johns Hopkins University, Baltimore, MD, United States of America, 4 Division of Biological Sciences, University of Washington, Bothell, WA, United States of America, 5 Department of Viticulture and Enology, University of California, Davis, Davis, CA, United States of America, 6 Department of Biology, University of New Mexico, Albuquerque, NM, United States of America, 7 Ecology, Evolution and Behavior Department, University of Minnesota, St. Paul, MN, United States of America, 8 Department of Forest Resources, University of Minnesota, Minneapolis, MN, United States of America, 9 Institute for Global Change Biology and School for Environment and Sustainability, University of Michigan, Ann Arbor, MI, United States of America, 10 Hawkesbury Institute for the Environment, Western Sydney University, Penrith South, NSW, Australia, 11 Department of Biology, University of North Carolina at Greensboro, Greensboro, NC, United States of America

⊕ These authors contributed equally to this work.
* tohlert@colostate.edu

This is a Registered Report and may have an associated publication; please check the article page on the journal site for any related articles.

## Abstract

The use of trait-based approaches to understand ecological communities has increased in the past two decades because of their promise to preserve more information about community structure than taxonomic methods and their potential to connect community responses to subsequent effects of ecosystem functioning. Though trait-based approaches are a powerful tool for describing ecological communities, many important properties of commonly-used trait metrics remain unexamined. Previous work with simulated communities and trait distributions shows sensitivity of functional diversity measures to the number and correlation of traits used to calculate them, but these relationships have yet to be studied in actual plant communities with a realistic distribution of trait values, ecologically meaningful covariation of traits, and a realistic number of traits available for analysis. To address this gap, we used data from six grassland plant communities in Minnesota and New Mexico, USA to test how the number of traits and the correlation between traits used in the calculation of eight functional diversity indices impact the magnitude of functional diversity metrics in real plant communities. We found that most metrics were sensitive to the number of traits used to calculate them, but functional dispersion (FDis), kernel density estimation dispersion (KDE dispersion), and Rao's quadratic entropy (Rao's Q) maintained consistent rankings of communities across the range of trait numbers. Despite sensitivity of metrics to trait correlation, there was no consistent pattern between communities as to how metrics were affected by the correlation of traits used to calculate them. We recommend that future use of evenness metrics include sensitivity analyses to ensure results are robust to the number of traits used

**Data Availability Statement:** All data and scripts may be found at https://github.com/kaitkimmel/FDiv/tree/master/R.

**Funding:** NSF DEB-1257965 (Kenneth D. Whitney); NSF DBI- 1725683, NSF DEB-1753859, NSF DEB-1831944 (Sarah E. Hobbie); NSF DBI- 1725683, NSF DEB-1753859, NSF-DBI-2021898 (Peter B. Reich), NSF DEB-0841917 (Elisabeth Forrestel). The funders had no role in study design, data collection and analysis, decision to publish, or preparation of the manuscript.

**Competing interests:** The authors have declared that no competing interests exist.

to calculate them. In addition, we recommend use of FDis, KDE dispersion, and Rao's Q when ecologically applicable due to their ability to produce consistent rankings among communities across a range of the numbers of traits used to calculate them.

## Introduction

Trait-based diversity measures have advanced the field of community ecology by increasing our understanding of both community assembly and diversity impacts on ecosystem functions [1, 2]. Functional diversity metrics allow researchers to quantify multiple facets of diversity, place an emphasis on mechanisms of community assembly, and provide a 'common currency' by which communities can be compared across sites and ecosystems [3, 4].Traditional measures for characterizing communities, such as species richness and species ordinations, use species' taxonomic classifications as discrete units, but functional diversity metrics can preserve more information about community assembly and function by including traits of species organized on continuous axes [5, 6].

Several aspects of functional and taxonomic diversity have been extensively studied. Scientists have probed functional diversity's correlation with species richness [7, 8] and ecosystem functioning [4], the importance of intraspecific trait variation for diversity [4, 9, 10], and the ecological hypotheses that functional diversity metrics can test, such as optimal strategies or functional turnover [6, 11]. Many taxonomic measures of community diversity have been extensively studied for their mathematical properties to allow these metrics to be comparable across sites and ecosystems, such as Shannon's diversity and Simpson's evenness that have mathematical characteristics linked to species number [12, 13]. Similarly, functional diversity metrics have mathematical characteristics that may cause the number or type of traits used to calculate the metric to impact the measure. For example, multidimensional metrics are calculated with additional dimensions for each additional trait included, and the correlation between traits affects the importance of each dimension to the metric [14]. Therefore, functional diversity could differ among replicate plots or sites simply because of the number or types of traits used to calculate the metric without any underlying ecological basis. Though single-trait indices are an effective tool for linking trait diversity to specific ecosystem processes [15, 16], indices based on multiple traits may better match ecological theories of community assembly around multidimensional niche space [17–19]. As use of multi-trait functional diversity increases, it is important to determine the conditions under which they reflect ecological processes as opposed to mathematical patterns.

Studies using simulated communities have tested whether the number and correlation of traits used in functional diversity metrics can impact the magnitude of the metric [7, 20]. Using simulated data, Legras et al. [20] showed that functional richness and functional divergence metrics decreased with increased trait number, but functional evenness metrics were not responsive to increasing trait numbers. Also using simulated data, Cornwell et al. [7] showed that convex hull volume (commonly referred to as "functional richness") tended to decrease with increasing correlation among traits included in the metric calculation, and that the decrease was greater in more species-rich communities. The limitations of functional diversity metrics described in these studies with simulated community data could be exacerbated when applied in natural communities. Calculating functional diversity measures in natural communities poses additional challenges both ecological and practical. Real plant communities are non-random assemblages of species which are influenced by competitive

interactions, coexistence, mutualisms, niche partitioning, and environmental filtering among many other processes of community assembly [21–26]. Functional diversity metrics are likely to exhibit patterns due to ecologically meaningful correlation of traits in real communities, in particular, among suites of traits typically used in community ecology such as the leaf economic spectrum and root economic spectrum [27, 28]. Moreover, real data collection introduces constraints on trait data, such as realistic numbers of traits collected given limited resources and missing trait data, particularly for rare species. Functional diversity metrics, therefore, are most often calculated with fewer traits and fewer species than those in studies based on simulated communities.

The field lacks clear guidelines for researchers to follow when choosing the number and types of traits to include when calculating functional diversity metrics. Decisions are often based on researcher intuition and the practices of similar studies, but such intuition and interpretation of trait selection can be improved by rigorous exploration of the impact of trait selection on diversity metrics [4, 29, 30]. These decisions can fall along a spectrum of options ranging from selecting the minimum number of traits needed to calculate a metric to using every trait available. For example, some studies suggest that researchers use a small number of traits related to certain ecosystem properties or other topics of interest (e.g., [8]), regardless of how correlated they may be. Other studies use all available traits in order to maximize the dimensions of diversity being studied in an effort to comprehensively assess the niche space that species and communities occupy (e.g., [31]). Choosing traits that are highly correlated can result in an underrepresentation of the diversity of functions present by overemphasizing groups of traits which describe similar processes, such as traits involved in the leaf economics spectrum [32]. Further, functional diversity metric calculation in high dimensional space can require dimensionality reduction–another decision that can impact the metrics. However, few studies scrutinize how these decisions can impact conclusions when using functional diversity metrics to characterize communities.

Here, we aimed to understand how the number of traits and correlation between traits impact functional diversity values. We focused on eight measures of functional diversity that express principal facets of community trait composition (see Table 1 for more details on each metric): functional richness (FRich), functional evenness (FEve), functional divergence (FDiv), functional dispersion (FDis), Rao's quadratic entropy (Rao's Q), kernel density estimation (KDE) richness, KDE evenness, and KDE dispersion [33–35]. We used trait data from real (natural/intact and experimental) plant communities, which allowed us to understand how these metrics respond to a realistic spread of traits and species richness. In this study, we used trait data collected from six U.S. grassland communities at two sites to test impacts of trait number and identity in functional diversity metric values. Our dataset included plant traits collected on location at these sites that include both naturally assembled and planted communities.

Specifically, we asked:

(1) Do functional diversity metrics exhibit specific patterns with respect to the number and correlation of traits used? Based on findings from [20], we expect functional richness, KDE richness, functional dispersion, and functional divergence to decrease with increasing numbers of traits, but for Rao's Q to increase [36] and functional evenness to be unresponsive to the number of traits. We do not have a priori hypotheses for KDE evenness and KDE dispersion since properties of these metrics have yet to be explicitly studied. Based on [7], we expect that functional richness will be greater when traits are less correlated. However, we do not have directional hypotheses with respect to effects of trait correlation on the rest of the metrics.

**Table 1. Description of tested trait metrics and examples of usage.**

| Functional diversity metric | Abbreviation | Ecological relevance | Examples of usage | Citations |
|---|---|---|---|---|
| Functional richness | FRich | Functional space filled by the community | De Vries and Bardgett 2016 [54] De la Riva et al. 2018 [55] Lourenco Jr. et al. 2021 [56] | Cornwell et al. 2006 [7], Villéger et al. 2008 [8] |
| Kernel density richness | KDE richness | Functional space filled by the community | Soares et al. 2022 [57] Piano et al. 2020 [58] Pavlek & Mammola 2021 [59] | Blonder 2018 [14], Mammola and Cardoso 2020 [35] |
| Functional evenness | FEve | The similarity trait abundances within the community | Bello et al. 2013 [60] Niu et al. 2016 [61] Biswas et al. 2019 [62] | Villéger et al. 2008 [8] |
| Kernel density evenness | KDE evenness | Similarity of trait abundances within the community | Soares et al. 2022 [57] Piano et al. 2020 [58] | Mammola and Cardoso 2020 [35] |
| Functional dispersion | FDis | Average trait difference between individuals within the community | Zuo et al. 2021 [63] Shovon et al. 2020 [64] Griffin-Nolan et al. 2019 [53] | Laliberte and Legendre 2010 [34] |
| Functional divergence | FDiv | Average trait difference between individuals within the community | Jäschke et al. 2020 [65] Ebeling et al. 2018 [66] Thakur & Chawla 2019 [67] | Villéger et al. 2008 [8] |
| Rao's quadratic entropy | Rao's Q | Average trait difference between individuals within the community | De Bello et al. 2009 [68] Ebeling et al. 2014 [69] Pillar et al. 2013 [70] Wang et al. 2018 [71] | Rao 1982 [73], Botta-Dukát 2005 [47] |
| Kernel density dispersion | KDE dispersion | Average trait difference between individuals within the community | Piano et al. 2020 [58] Greenop et al. 2021 [72] | Mammola and Cardoso 2020 [35] |

(2) Is metric sensitivity to trait number/type consistent across sites and experiments? If metric sensitivity is consistent across sites, it will be easier to standardize functional diversity metrics across different studies. If sensitivity is not consistent across sites, further caution will be needed in interpreting cross-site comparisons of functional diversity.

## Methods

We performed methods as described in the Registered Report Protocol [37]. Alterations to the protocol are explained in Table 2 and the following methods represent only those methods performed for this study.

### Site descriptions

Here we used data from six communities in two United States grasslands that span a range of species diversity. Two communities were from a site with natural species assemblages and four communities were from a site with planted species assemblages in order to be representative of the state of grassland studies where some use naturally assembled communities while others use planted communities. Cedar Creek Ecosystem Science Reserve (CDR; East Bethel, Minnesota, USA; latitude = 45.4, longitude = -93.2) is in central Minnesota and classified as a tallgrass prairie. According to Koppen and Geiger classification, the climate is characterized as cold continental with hot summer, but without a dry season [38]. The mean growing season (May–August) precipitation is approximately 420 mm, mean minimum growing season temperature is 12°C, and mean maximum growing season temperature is 25°C (1982–2016

**Table 2. Summary of methodological changes from registered report doi.org/10.1371/journal.pone.0272791.**

| Proposed method | Used method | Rationale for change |
|---|---|---|
| Data sources from Cedar Creek, Konza Prairie, and Sevilleta | Data sources from only Cedar Creek and Sevilleta | Trait data from Konza Prairie did not meet our 80% threshold of coverage of the community as stated in the registered report and therefore was not used. Specifically, the annually burned community had a maximum trait coverage of 77%, the annually burned and grazed community has a maximum trait coverage of 56%, the community burned every 20 years had a maximum trait coverage of 33%, and the community burned every 20 years that was grazed had a maximum trait coverage of 34%. |
| At Cedar Creek, trait data come from the monoculture plots of the BioCON experiment that correspond to the $CO_2$ and N treatments to match with 16-species community plots | Trait values for select species were pulled from an adjacent experiment to ensure total trait coverage. | SLA was not available for two species (*Poa pratensis* and *Bouteloua gracilis*). Seed mass was not available for five species (*Achillea millefolium*, *Amorpha canescens*, *Anemone cylindrica*, *Asclepias tuberosa*, and *Petalostemum villosum*). These two traits were calculated from the Big Biodiversity experiment trait dataset and substituted for all $CO_2$ and N communities. |
| | | Root %C and %N was only available for *Anemone cylindrica* in one of the CO2 and N communities in monoculture. The other communities were filled in with this value. |
| Use ten traits for Cedar Creek | Used nine traits for Cedar Creek | We miscounted the number of traits available. The registered report only named nine traits to be used. |
| Compare linear, quadratic, cubic, and quartic models | Compare null, linear, and quadratic models | Many of the best-fitting linear models had a slope close to 0. We decided to add the intercept-only null model to be able to interpret between linear models that had a slope and those with slopes close to 0. The null model indicates that the independent variables did not describe the variation of the response variable. |
| | | Cubic and quartic models were initially included to mimic the methods of Legras et al. (2020). We did not use cubic and quartic fits as we could not interpret the meaning of these higher-order model fits as it relates to the use of these indices. |
| Perform correction for multiple comparisons | No correction for multiple comparisons | We are not focusing on the p-values of our best-fit lines, rather just comparing model fits. We only used p-values to determine the best fit of a line and not in the traditional sense of determining whether a certain variable was statistically significant. Therefore, there was no need to use multiple comparison corrections in our analysis. |
| Sensitivity analysis of PCoa | No sensitivity analysis | We tried to run an initial sensitivity analysis calculating FRic with different *m* values—the parameter in the FDiv package that sets the number of dimensions. However, dimensions were automatically reduced to two since you cannot have more axes than species and our data have numerous plots with very few species. Therefore, we could not perform a full sensitivity analysis. |

period; http://www.cedarcreek.umn.edu/research/data). Soils at Cedar Creek are characterized as nutrient-poor entisols derived from a glacial outwash sand plain [38]. The study from Cedar Creek consists of artificially planted communities. The Sevilleta National Wildlife Refuge (SEV) is in central New Mexico at the northern edge of the Chihuahuan Desert (latitude = 34.4, longitude = -106.7). The Sevilleta includes desert grasslands, and the climate is characterized as cold semi-arid according to the Koppen and Geiger classification [38]. The growing season is characterized by two rainy periods (March—May and July—September) split by a dry period. The mean monsoon growing season precipitation is approximately 150 mm and the mean monsoon growing season temperature is 22˚C.

## Community composition data

We used community composition data from two communities at the Sevilleta and four at Cedar Creek (n = 6 communities total) collected within a single year (2018 for Sevilleta, 2020 for Cedar Creek) to characterize the functional diversity of grassland plant communities.

At Cedar Creek, we used community composition data from all 16-species plots in a biodiversity, $CO_2$, and nitrogen addition experiment (BioCON, n = 48; 12 plots for each $CO_2$-N combination). All 16-species plots were originally planted with the same mixture of species (*Achillea millefolium*, *Amorpha canescens*, *Andropogon gerardii*, *Anemone cylindrica*, *Asclepias tuberosa*, *Bouteloua gracilis*, *Bromus inermis*, *Elymus repens*, *Koeleria cristata*, *Lespedeza capitata*, *Lupinus perennis*, *Petalostemum villosum*, *Poa pratensis*, *Schizachyrium scoparium*, *Solidago rigida*, and *Sorghastrum nutans*) such that all species were seeded at the same density in 1997. Plots were weeded every year to remove invading species. Through time, the plots can lose species (and regain those) but could never gain new species. Further, species abundances shifted from the equal proportion planted in the first year. Every August, species abundances were visually estimated in a 1 $m^2$ permanent plot.

At Sevilleta, we used community composition data from two observational sites, one in a Great Plains grassland ecosystem and the other in a desert grassland ecosystem. The Great Plains grassland is dominated by *Bouteloua gracilis* (blue grama), a long-lived, caespitose, C4 perennial grass common throughout much of the United States and Canada. The desert grassland is dominated by *B. eriopoda* (black grama), a stoloniferous C4 perennial grass common in the southwestern United States and Mexico. These two dominant perennial grasses account for about 80% of vegetative cover in their respective ecosystems. Each site has 28 1 $m^2$ quadrats which were sampled in September of 2018, at the peak of the post-monsoon growing season. In each quadrat, plants were identified to species and their percent ground cover was visually estimated.

## Trait data

Trait data were collected on individuals found at each of the different sites. Thus, our trait data are representative of the traits actually found in the given community and not just an average independent of location. Traits include measurements from leaves (e.g. specific leaf area), stems (e.g. stem dry matter content), roots (e.g. root dry matter content), whole-plant (e.g. height), and ecological attributes (e.g. amount of nitrogen in monoculture). Including traits across these measurement categories provides a more-complete representation of community assemblages [39–42]. For detailed descriptions of trait collection protocols at each site, see the S1 and S2 Figs.

At Cedar Creek, we used trait data collected in the monoculture plots of the BioCON experiment that correspond to the $CO_2$ and N treatments to match with 16-species community plots. Trait data were collected between 1998 and 2020. Some traits were collected over multiple years whereas others were only collected once. In total, there were 9 distinct traits: specific leaf area (SLA), I* (the amount of light at the soil surface in monoculture), R* (the amount of nitrogen in monoculture), root %C, root %N, total root biomass, shoot %N, shoot %C, and seed mass.

At Sevilleta, we used trait data collected from 2017–2021 on individuals growing under ambient conditions near permanent ambient plots used to monitor plant communities. The full suite of traits were often measured on the same individuals, up to 10 individuals per species. In total there were 10 distinct traits: maximum plant height, leaf dry matter content, specific leaf area, d15N, d13C, leaf %N, leaf %C, stem dry matter content, root dry matter content, and photosynthetic pathway.

For each trait at each site, we calculated an average trait value based on all the measurements for the given species and trait. We acknowledge that this obscures variation within a given trait (intraspecific variation) for a species; such variation can be quite important for some questions [43–46]. The impacts of intraspecific variation in this study are minimized by only using trait values collected at each site, but sufficient data were not collected for each trait of each species to include intraspecific variation into our analysis. Before analysis, we removed species that had less than 100% trait coverage. We made sure that the communities were still represented by at least 80% of species abundance–this approach de-emphasizes the importance of rare species, but is a logistical constraint faced by many researchers doing trait analyses. This ensured that we represented the community to the best of our ability with the given trait data.

## Brief background on functional diversity metrics

We focused our analyses on eight common functional diversity metrics: functional richness (FRich) [8], functional evenness (FEve) [9], functional dispersion (FDis), functional divergence (FDiv), Rao's quadratic entropy (Rao's Q), kernel density estimation (KDE) richness, KDE evenness, and KDE dispersion [34]. FRich is the multidimensional equivalent of a range [8]. It is calculated as the convex hull volume that is made from all trait values for up to $n$ traits in the community. The number of dimensions used to calculate the final volume can be reduced from the total trait number [45]. FEve is the minimum spanning tree to quantify the regularity of branch lengths and the evenness in trait relative abundances. For each branch, $l$, of the minimum spanning tree, the weighted evenness (EW) is calculated as $EW_l = \frac{dist(i,j)}{w_i+w_j}$ where $i$ and $j$ are species, and $w_i$ is the relative abundance of species $i$. Then, the partial weighted evenness ($PEW$) is calculated for each branch as $PEW_l = \frac{EW_l}{\sum_{l=1}^{S-1} EW_l}$, where $S$ is the total number of species in the community. FEve is then defined as $\frac{\sum_{l=1}^{S-1} \min\left(PEW_l, \frac{1}{S-1}\right) - \frac{1}{S-1}}{1 - \frac{1}{S-1}}$ [8]. FDis is the weighted mean distance between species and a weighted-centroid. It is calculated as $\frac{\sum a_j z_j}{\sum a_j}$ where $a_j$ is the relative abundance of species $j$ and $z_j$ is the distance species $j$ is from the weighted centroid [34]. FDiv is a relative abundance-weighted spread of traits along a trait axis independent of functional richness and is calculated as $\frac{\Delta d + \bar{dG}}{\Delta|d| + \bar{dG}}$ where $\bar{dG}$ is the mean distance of species to the weighted-centroid and $\Delta d$ is the sum of relative abundance-weighted deviances from the weighted-centroid [9]. Rao's Q measures the pairwise differences in traits between species in a community and is calculated as $\sum_{i-1}^{s-1} \sum_{j=i+1}^{S} d_{ij} p_i$ where $S$ is the number of species in the community, $d_{ij}$ is the functional difference between the $i$-th and $j$-th species, and $p$ is a vector of relative abundance values [46]. These five functional diversity metrics commonly incorporate distance measures by reducing dimensionality using principal coordinates analysis (PCoA) to return PCoA axes which are used to calculate the functional diversity metrics. However, we will avoid this dimensionality reduction for all metrics except FRich, see discussion in Functional Diversity Calculations section. $n$-dimensional hypervolumes use Gaussian kernel density estimation (KDE) to create a relative abundance-weighted probability distribution of traits in multidimensional space [35]. KDE richness is the total volume of the n-dimensional hypervolume created from unweighted trait values present in the community. KDE evenness is the overlap between the abundance-weighted $n$-dimensional hypervolume and a similar hypervolume in which all traits and abundances are distributed evenly. KDE dispersion is the average distance between random points within the n-dimensional hypervolume and the hypervolume centroid.

## Functional diversity calculations

For each site, we followed the same protocol for calculating functional diversity metrics. We calculated FRich, FEve, and FDis, FDiv, and Rao's Q using the 'FD' package in R [46] using both Gower and Euclidean dissimilarity as the distance measure, along with using the hypervolume package in R to calculate KDE $n$-dimensional hypervolumes which are passed to the 'bat' package to create KDE richness, KDE evenness, and KDE dispersion [35, 48]. Functional diversity metrics from the 'FD' package and kernel density estimation are among the most-used metrics for quantifying trait-based diversity within communities due to both ease of use and ecological relevance [35, 45]. To understand the impact of trait number on functional diversity, each functional diversity metric was calculated using all possible combinations of two traits up to all possible combinations of the maximum number of traits at each site. For example, at Sevilleta there are 10 different traits so there are 45 2-trait calculations, 120 3-trait calculations, 210 4-trait calculations, and so forth up to 10 9-trait calculations and 1 10-trait calculation. This allows us to focus on the impact of trait number independent of the constituent set of traits used to calculate the metric.

To calculate the five metrics using the 'FD' package, we first calculated a species-trait distance matrix using both Gower (categorical and continuous traits) and Euclidean (continuous traits only) distances. These distance matrices were calculated with both scaled and centered and non-scaled trait data for each community. Centering was done by subtracting the trait mean from each observation and scaling was done by dividing the centered traits by their standard deviations (as in the 'FD' package). These distance matrices along with a species-abundance matrix are the input for the 'FD' package. The 'FD' package performs a principal components analysis on the full species-trait distance matrix. Dimensionality reduction only occurs for FRic and FDiv metric calculation. For all FRich and FDiv analyses, we hold the number of dimensions equal to 2, similar to Legras et al. [20]. Because some communities only had two species, we did not perform a sensitivity analysis to look at how increased dimensionality impacted our results since these species depauperate communities would be excluded. Further, when running the analyses for FRic, FEve, and FDiv, one plot from each Sevilleta community with only one species present was removed because FRic, FEve, and FDiv are undefined in monoculture communities. For calculation of KDE metrics, a species-abundance matrix and a species-trait matrix were loaded for each of the six communities while the distance matrix was set to either Gower or Euclidean depending on the calculation being performed. To measure the effects of trait correlation on functional diversity, we focused on metrics calculated with four traits only to standardize between sites. We calculated the minimum, maximum, and mean correlation between the traits at each community. Only combinations of four traits were used as a balance between reduction of noise in the evaluation of minimum and maximum correlation (calculated as the pairwise correlation of just two traits) and the lower end of the number of traits likely to be used to calculate these metrics in the literature.

## Statistical analyses

For each community separately, we ran mixed effects models to test the dependence of the eight functional trait metrics on trait number and on trait correlation using the lme function from the 'nlme' package in R [48]. To examine how trait number impacts the values of a given functional trait metric, we ran the model Metric ~ trait number for 2–10 unique traits. To examine how trait-trait correlation impacts the values of a given functional diversity metric, calculated three metrics of trait correlation for each unique combination of four traits: 1) min trait correlation is the minimum two-trait correlation among the set of four traits, 2) max trait

correlation is the maximum two-trait correlation among the set of four traits, and 3) mean trait correlation is the average of all trait-trait correlations among the set of four traits. Next, we ran three models for each community: Metric ~ min trait correlation, Metric ~ max trait correlation, Metric ~ mean trait correlation. We explored which functional form of the predictor variables best fit the spread of the functional metric data by fitting null (e.g., intercept only), linear, and quadratic fits. We selected models based on best fit using AIC values. We accounted for repeated samples within plots by fitting plot as a random effect and using an autoregressive correlation structure. Raw model outputs for the total 1,152 models are generated by scripts n_traitStats_simplified.R, n_traitStats_euc_simplified.R, max_corrStats_simplified.R, max_corrStats_euc_simplified.R,min_corrStats_simplified.R, min_corrStats_euc_simplified.R, mean_corrStats_simplified.R, mean_corrStats_euc_simplified.R' in this GitHub repository <https://github.com/kaitkimmel/FDiv/tree/master/R>.

## Results

### Data processing summary

Cedar Creek and Sevilleta had adequate trait coverage to proceed with the analyses. At Cedar Creek, all plots from all communities had 100% trait coverage. Overall, we included data from four different communities (defined as different nitrogen fertilization and carbon dioxide enrichment treatments) each with 16 total plots. At Sevilleta, the minimum plot-level trait coverage from the 'Blue grama' communities was 79.96%. We included all the plots as 79.96% was close to the 80% trait-coverage threshold. Thus, we had 28 individual plots within this community. The minimum plot-level trait coverage from the 'Black grama' communities at Sevilleta was 77.72%. We removed this one plot and had a total of 27 plot from the 'Black grama' community.

### Sensitivity of functional diversity metrics to trait number

FDis calculated with Gower dissimilarity was insensitive to the number of traits used to calculate it across all six communities (Fig 1I; Table 3). KDE richness, KDE dispersion, and Rao's Q with Gower dissimilarity were negatively correlated with the number of traits (Fig 1B, 1J and 1M). However, the rankings of communities from low to high values remained consistent. That is, community order was maintained within these three metrics across the range of trait numbers. Similarly, FRich, KDE richness, FDis, KDE dispersion, and Rao's Q calculated with Euclidean dissimilarity maintained rankings among communities throughout the range of trait numbers, though these metrics all increased with the number of traits (Fig 1C, 1D, 1K, 1L and 1O; Table 3). For both Gower and Euclidean dissimilarity, FEve, KDE evenness, and FDiv had different rankings among communities depending upon the number of traits used to calculate them.

### Sensitivity of functional diversity metrics to trait correlations

Trait correlations (mean, maximum, and minimum trait-trait correlation for combinations of four traits) had limited power to predict the calculated metrics for both dissimilarity matrices and the relationships between metrics and trait correlation varied widely among communities (Fig 2, S1 and S2 Figs). Consequently, we observed inconsistent rankings among communities with respect to trait correlation (Fig 2, S1 and S2 Figs). For example, for FRich calculated with an Euclidean matrix at maximum correlation of 0.40, FRich of the CDR2 community was greater than that of CDR1, whereas at maximum correlation of 0.75, the reverse was true, FRich of CDR1 was greater than that of CDR2 (Fig 2C). Such reorganizations of community rankings were common across functional diversity metrics, distance matrices, and correlation metrics (max, min, mean) (Fig 2). In some cases, metrics were not responsive to trait

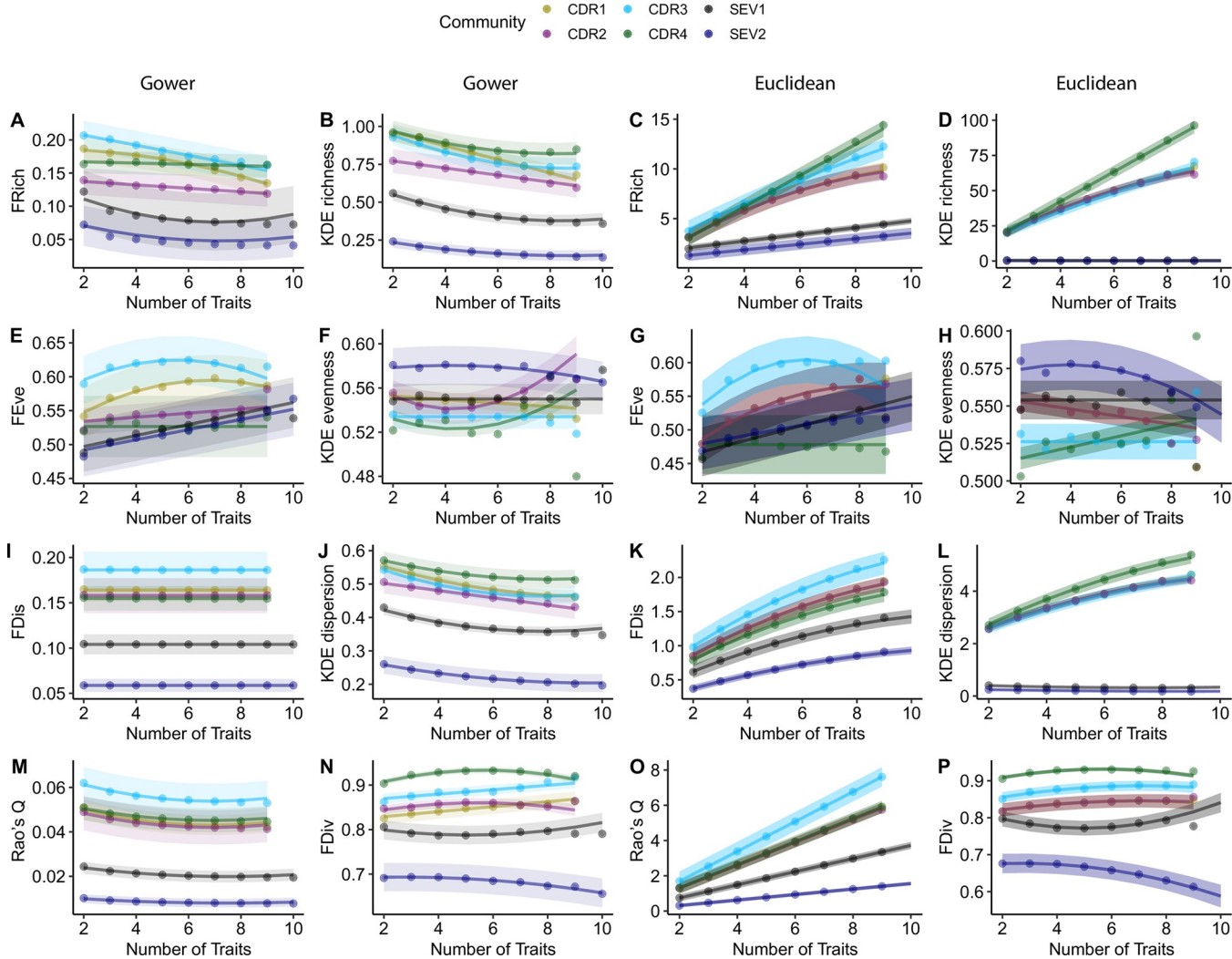

**Fig 1. The relationship between trait number and functional diversity metrics using both Gower (columns 1 & 2) and Euclidean (columns 3 & 4) dissimilarity matrices.** Each point represents the mean value for the given community for a certain number of traits used to calculate the metric. Solid lines are the predicted fits of the best model and shaded regions are +/- SE of the predicted fit. Different colors represent the six communities used in this study (four experimental communities at Cedar Creek Ecosystem Science Reserve, CDR, and two natural communities at Sevilleta National Wildlife Refuge, SEV). N = 6,024 observations for each Cedar Creek community (432 2-trait combinations, 1,008 3-trait combinations, 1,512 4-trait, 1,512 5 trait combinations, 1,008 6-trait combinations, 432 7-trait combinations, 108 8-trait combinations, 1 9-trait combination); n = 27,351 observations for SEV1; n = 28,364 observations for SEV2.

correlations (e.g., null models were the best fit for 19% of all max correlation models, 20% of min correlation models, 18% of mean correlation models).

## Distance matrices

Overall, metrics calculated with Gower and Euclidean distance matrices maintained consistent results with respect to the ranking of the six communities (four experimental planted communities from Cedar Creek and two natural communities at Sevilleta) for different metrics. The Euclidean distance matrix tended to amplify the differences among communities as the numbers of traits increased for KDE richness, FDis, Rao's Q, and KDE dispersion. Certain functional diversity metrics were poor at maintaining the ranking of communities across ranges of

**Table 3. Counts of the number of best fit models for different predictor variables and functional diversity metrics calculated using Gower and Euclidean distances.** For six communities, each of eight metrics was calculated across a range of the four predictor variables using both Gower and Euclidean dissimilarity matrices. Three functional forms of models were tested: intercept only, linear, and quadratic. In this table, counts underneath those functional form columns display the number of communities (out of six) for which that functional form was the best model. For example, FRich predicted by trait number and calculated with Gower dissimilarity was best predicted by a linear model for three communities and best predicted by a quadratic model for three communities. In total, this table summarizes the results of 384 models.

| Predictor | Metric | Intercept Only | | Linear | | Quadratic | |
|---|---|---|---|---|---|---|---|
| | | Gow | Euc | Gow | Euc | Gow | Euc |
| Trait Number | FRich | 0 | 0 | 3 | 3 | 3 | 3 |
| | KDE richness | 0 | 0 | 2 | 1 | 4 | 5 |
| | FEve | 1 | 1 | 3 | 2 | 2 | 3 |
| | KDE evenness | 2 | 2 | 1 | 3 | 3 | 1 |
| | FDis | 6 | 0 | 0 | 0 | 0 | 6 |
| | KDE dispersion | 0 | 0 | 1 | 0 | 5 | 6 |
| | FDiv | 0 | 0 | 2 | 0 | 4 | 6 |
| | Rao's Q | 0 | 0 | 0 | 6 | 6 | 0 |
| Maximum Correlation | FRich | 0 | 1 | 2 | 1 | 4 | 4 |
| | KDE richness | 1 | 0 | 2 | 2 | 3 | 4 |
| | FEve | 4 | 1 | 1 | 2 | 1 | 3 |
| | KDE evenness | 2 | 2 | 1 | 3 | 3 | 1 |
| | FDis | 1 | 0 | 0 | 1 | 5 | 5 |
| | KDE dispersion | 1 | 1 | 2 | 1 | 3 | 4 |
| | FDiv | 1 | 2 | 3 | 2 | 2 | 2 |
| | Rao's Q | 0 | 1 | 2 | 2 | 4 | 3 |
| Minimum Correlation | FRich | 0 | 1 | 0 | 0 | 6 | 5 |
| | KDE richness | 1 | 0 | 0 | 1 | 5 | 5 |
| | FEve | 3 | 4 | 3 | 1 | 0 | 4 |
| | KDE evenness | 1 | 2 | 1 | 2 | 4 | 2 |
| | FDis | 0 | 1 | 1 | 0 | 5 | 5 |
| | KDE dispersion | 1 | 0 | 2 | 3 | 3 | 3 |
| | FDiv | 3 | 2 | 1 | 1 | 2 | 3 |
| | Rao's Q | 0 | 0 | 1 | 2 | 5 | 4 |
| Mean Correlation | FRich | 1 | 0 | 1 | 0 | 4 | 6 |
| | KDE richness | 1 | 1 | 1 | 0 | 4 | 5 |
| | FEve | 2 | 1 | 3 | 1 | 1 | 4 |
| | KDE evenness | 3 | 3 | 1 | 0 | 2 | 3 |
| | FDis | 1 | 0 | 2 | 1 | 3 | 5 |
| | KDE dispersion | 2 | 1 | 0 | 0 | 4 | 5 |
| | FDiv | 1 | 0 | 0 | 2 | 5 | 4 |
| | Rao's Q | 0 | 0 | 4 | 2 | 2 | 4 |

the number and correlation of traits (e.g. FEve; Figs 1E, 1G, 2E and 2G). However, for instances in which the ranking of communities changed, neither Gower nor Euclidean distance improved the issues.

## Discussion

### Sensitivity to the number of traits

Our study aimed to understand the sensitivity of functional diversity metrics to trait inputs. Here, we found that FDis had consistent values when calculated with Gower dissimilarity such

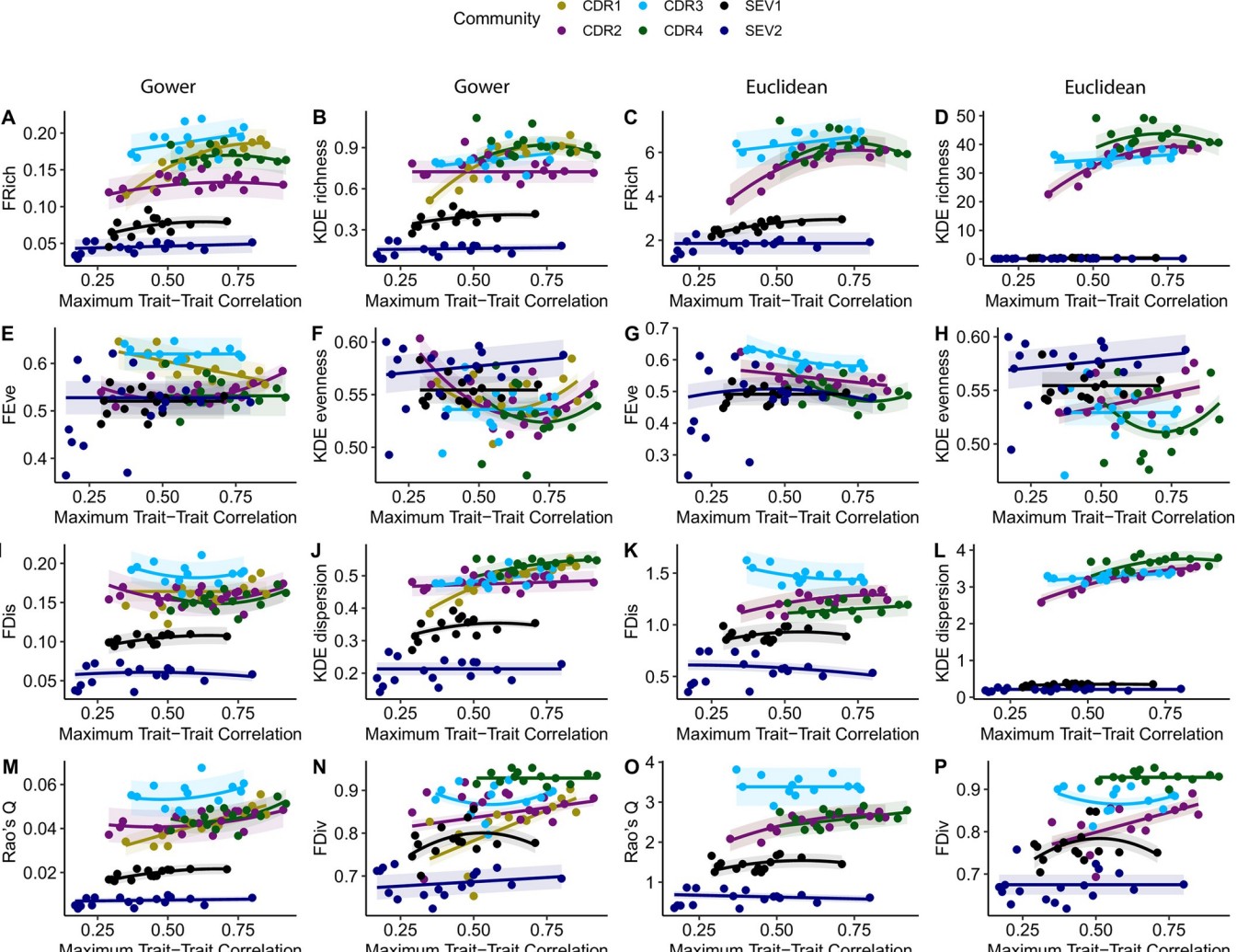

**Fig 2. The relationship between the maximum trait-trait correlation for each set of 4 traits and functional diversity metrics using both Gower (columns 1 & 2) and Euclidean (columns 3 & 4) dissimilarity matrices.** Each point represents the mean value for the given community at that correlation. Solid lines are the predicted fits of the best model and shaded regions are +/- SE of the predicted fit. Different colors represent the six communities used in this study (four experimental communities at Cedar Creek Ecosystem Science Reserve, CDR, and two natural communities at Sevilleta National Wildlife Refuge, SEV). N = 1,512 observations for each CDR community; n = 3,402 for SEV1; n = 3,528 for SEV2.

that, across all communities, there was no correlation with the number of traits used (e.g., it remained consistent regardless of trait number). This suggests that FDis provides reliable values across different communities and sets of traits, making it a potentially valuable tool for assessing patterns of functional diversity across communities and ecosystems. Similarly, Rao's Q, KDE dispersion, and KDE richness maintained consistent ordered rankings of metrics among communities across the range of trait numbers for both Gower and Euclidean dissimilarity matrices along with FDis calculated with Euclidean dissimilarity. Other metrics (FEve, KDE evenness, FDiv) had less consistency across the range of traits used to calculate them as relative rankings of different communities changed with the number of traits used in constructing the metrics.

A previous simulation study found no sensitivity to the number of traits for FEve and FDiv and magnitude decreases with increasing trait number for FRich, KDE richness, FDis, and

Rao's Q calculated with Gower dissimilarity [20]. Our results were consistent with Legras et al. [20] for Rao's Q and KDE richness, but we found FDis unresponsive to the number of traits and FDiv, FEve, and KDE evenness to have inconsistent slopes among communities. These differences in findings might be attributed to the inherent complexity and noise present in real-world data. Real communities often contain anomalous species with outlier trait values (e.g. a gymnosperm among angiosperms, a tree seedling among herbaceous plants, or other rare species outlier values), which can exert considerable influence on evenness indices.

We found further discrepancies with previous studies reporting results using Euclidean dissimilarity. Previous studies found no sensitivity of FEve and FDiv [20, 36], and increases with increasing trait number for KDE richness, Rao's Q [35], and FRich [20] when calculated with Euclidean dissimilarity. Our results of increasing FDis with the number of traits matches Legras et al. [20] while Zhang et al. [36] reported no sensitivity. A potential explanation for this discrepancy is the difference in the number of traits considered. Our study and the simulated data in Legras et al. [20] were limited to a maximum of 10 traits while Zhang et al. [36] used a maximum of 34 traits. While there may indeed be no sensitivity of FDis calculated with Euclidean dissimilarity at numbers of traits as high as 34, studies employing these metrics more often use fewer than 10 traits, within the range in which we found sensitivity.

Some metrics may be unreliable measures for comparing functional diversity among communities since comparisons are dependent upon the number of traits used to calculate them. Specifically, FRich, FEve, and KDE evenness showed crossing slopes among communities (i.e. ranking of communities changed with the number of traits) for both Gower and Euclidean distance matrices. The inconsistency of communities' relationships to one another across the range of the number of traits raises concerns. For example, community CDR4 had a greater FEve than community SEV1 when using four traits, but CDR4 had a smaller FEve than SEV1 when calculated with eight traits. Such discrepancies have the potential to introduce discordant results in the literature, even when otherwise identical studies have been conducted. This is particularly concerning given the often-arbitrary nature of selecting the number of traits used in a study. The number of traits is often dictated by the resources available to collect data or the completeness of publicly available data [49, 50]. Therefore, additional analyses are warranted when using these metrics to ensure that results are not merely an artifact of the number of traits used to calculate the metrics. Interestingly, the more consistent indices—such as KDE alpha, FDis, Rao's Q, and KDE dispersion—measure similar ecological properties (e.g., the range of traits expressed in the community) as FRich [6, 33, 35] and therefore, these indices could be substituted for FRich in analyses. However, evenness metrics, both FEve and KDE evenness, quantify a different type of ecological property, the relative homogeneity of traits within a community [35]. These indices do not have obvious substitutes for measuring these ecological properties among the metrics studied here.

## Trait correlation concerns in calculating metrics: Much ado about nothing?

We found inconsistent and null relationships between metrics and the correlation of traits used for their calculations. Despite suggestions in the literature that trait selection should minimize correlation [35], lower levels of trait-trait correlation did not result in more or less clear comparisons among communities. Though nonlinearity was prevalent in our analysis, most trait metrics demonstrated similarity across the entire range of maximum trait-trait correlation (i.e. the range of values was relatively small). Lefcheck et al. [29] utilized simulated data and reported insensitivity of Rao's Q, FEve, and FDis to trait correlation except at very high levels of correlation (Pearson's $|R| > 0.95$). However, they observed that FRich and FDiv decreased with trait correlation—a trend that was not evident in our study. Additionally, Lefcheck et al.

[29] noted that sensitivity to trait correlation became most apparent for FDiv and FRich when larger numbers of traits were considered, whereas we only tested combinations of four traits. Notably, Mammola and Cardoso [35] recommended avoiding the use of highly correlated traits (Pearson $|r| > = 0.8$) when calculating KDE metrics. Though we did not find substantial support for such a cutoff, our set of collected traits also had very few combinations with correlation above 0.8. Overall, our results show no consistent link between trait correlation and the values of functional diversity metrics.

## Gower and Euclidean dissimilarity

Both Gower and Euclidean distances performed similarly, though diversity metric values varied more with the number of traits under Euclidean distance. Metrics that preserved rankings among communities across the trait number and correlation gradients did so with both Gower and Euclidean matrices. The primary difference driving the use of these distance matrices in the literature is that unlike Euclidean distance, Gower distance can conveniently accommodate categorical data. Categorical traits, such as photosynthetic pathway, growth form, and nitrogen fixation capacity, are often easy to collect and more reliably scored than continuous traits. Moreover, trait databases typically have a great deal of missing data [50, 51] and categorical traits are more reliably gap-filled *ad-hoc* (e.g. growth form may be determined from a picture or nitrogen fixation capacity pulled from literature) than continuous traits. Given that both distance matrices performed similarly across our broad range of functional diversity indices, trait correlations, and trait numbers, there is no clear reason to favor use of a particular matrix other than the ability of Gower matrices to include categorical trait types.

## Ecological significance of methodology and recommendations

Perhaps most important is to center ecological significance and interpretation when choosing traits and metrics. Though many metrics produced consistent results with respect to the rankings of communities from two to ten traits, there are many sensible reasons to include more than 2 traits in order to capture more dimensions of diversity [43, 52]. Similarly, though we found no obvious difference of results when including highly correlated traits, this may not be license to include the maximum number of traits available in all circumstances. One example of responsible use of traits is found in Griffin-Nolan et al. [53] in which traits were used to assess plant community responses to drought. In this case, the analyses focused solely on hydrological traits, ignoring many other traits commonly used in the literature (e.g. seed mass), but the authors correctly emphasized the importance of choosing only traits relevant to the particular treatments and plant functions of interest. While use of the maximum number of traits available may be justifiable when seeking to quantify diversity defined broadly, we discourage inclusion of traits without ecological rationale.

Based on our findings, we recommend use of FDis, KDE dispersion, and/or Rao's Q in analyses of functional diversity as all of these measures provide consistent results among communities at all numbers of traits tested. Additionally, due to the inconsistency of evenness metrics with respect to community rankings, we strongly recommend that any use of FEve or KDE evenness metrics include supplemental analyses to test whether results are consistent with different numbers of traits used to calculate them. Surprisingly, we found no rationale to favor a particular distance matrix; we simply suggest that the number of traits used or correlation of traits need not be a consideration when choosing between Gower and Euclidean dissimilarity matrices. While functional diversity indices enrich the toolbox for exploring trait-based plant diversity, it remains important to ensure that our findings and inferences are

rooted primarily in ecological principles rather than being solely reflective of the metrics employed in assessing functional diversity.

## Supporting information

**S1 Fig. The relationship between mean trait-trait correlation for each set of 4 traits and functional diversity metrics using both Gower (columns 1 & 2) and Euclidean (columns 3 & 4) dissimilarity matrices.** Each point represents the mean value for the given community for a specific number of traits. Solid lines are the predicted fits of the best model and shaded regions are +/- SE of the predicted fit. Different colors represent the six communities used in this study (four experimental communities at Cedar Creek Ecosystem Science Reserve, CDR and two natural communities at Sevilleta National Wildlife Refuge, SEV). N = 1,512 observations for each CDR community; n = 3,402 for SEV1; n = 3,528 for SEV2.
(PDF)

**S2 Fig. The relationship between the minimum trait-trait correlation for each set of 4 traits and functional diversity metrics using both Gower (columns 1 & 2) and Euclidean (columns 3 & 4) dissimilarity matrices.** Each point represents the mean value for the given community at that correlation. Solid lines are the predicted fits of the best model and shaded regions are +/- SE of the predicted fit. Different colors represent the six communities used in this study (four experimental communities at Cedar Creek Ecosystem Science Reserve, CDR, and two natural communities at Sevilleta National Wildlife Refuge, SEV). N = 1,512 observations for each CDR community; n = 3,402 for SEV1; n = 3,528 for SEV2.
(PDF)

## Acknowledgments

Thank you to two anonymous reviewers for providing feedback that improved the quality of this research.

## Author Contributions

**Conceptualization:** Timothy Ohlert, Kaitlin Kimmel.

**Data curation:** Kaitlin Kimmel, Cynthia Chang, Elisabeth Forrestel, Benjamin P. Gerstner, Sarah E. Hobbie, Peter Reich, Kenneth D. Whitney, Kimberly Komatsu.

**Formal analysis:** Timothy Ohlert, Kaitlin Kimmel.

**Funding acquisition:** Sarah E. Hobbie, Peter Reich, Kenneth D. Whitney, Kimberly Komatsu.

**Investigation:** Kaitlin Kimmel.

**Methodology:** Timothy Ohlert, Kaitlin Kimmel.

**Project administration:** Timothy Ohlert, Kaitlin Kimmel.

**Supervision:** Meghan Avolio, Kimberly Komatsu.

**Visualization:** Timothy Ohlert, Kaitlin Kimmel.

**Writing – original draft:** Timothy Ohlert, Kaitlin Kimmel.

**Writing – review & editing:** Kaitlin Kimmel, Meghan Avolio, Cynthia Chang, Benjamin P. Gerstner, Sarah E. Hobbie, Peter Reich, Kenneth D. Whitney, Kimberly Komatsu.

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
