## [Decision Letter · Decision Letter 0]

5 Mar 2024

PONE-D-24-03515The impact of trait number and correlation on functional diversity metrics in real-world ecosystemsPLOS ONE

Dear Dr. Ohlert,

Thank you for submitting your manuscript to PLOS ONE. After careful consideration, we feel that it has merit but does not fully meet PLOS ONE’s publication criteria as it currently stands. Therefore, we invite you to submit a revised version of the manuscript that addresses the points raised during the review process.

I received the report from two independent reviewers, one of them underlines some important weaknesses of the manuscript. I have also revised the manuscript confirming that some methodological aspects should be better described as well as the aims assumptions.

We look forward to receiving your revised manuscript.

Kind regards,

Francesco Boscutti

Academic Editor

PLOS ONE

“NSF DEB-1257965 (Kenneth D. Whitney); NSF DBI- 1725683, NSF DEB-1753859, NSF DEB- 1831944 (Sarah E. Hobbie); NSF DBI- 1725683, NSF DEB-1753859, NSF-DBI-2021898 (Peter B. Reich), NSF DEB-0841917 (Elisabeth Forrestel).”

“We appreciate the funding sources that facilitated data collection and management for this project

including NSF DEB-1257965 (Kenneth D. Whitney); NSF DBI- 1725683, NSF DEB-1753859, NSF DEB-1831944 (Sarah E. Hobbie); NSF DBI- 1725683, NSF DEB-1753859, NSF-DBI-2021898 (Peter B. Reich), NSF DEB-0841917 (Elisabeth Forrestel).”

“NSF DEB-1257965 (Kenneth D. Whitney); NSF DBI- 1725683, NSF DEB-1753859, NSF DEB- 1831944 (Sarah E. Hobbie); NSF DBI- 1725683, NSF DEB-1753859, NSF-DBI-2021898 (Peter B. Reich), NSF DEB-0841917 (Elisabeth Forrestel).”

4. Please update your submission to use the PLOS LaTeX template. The template and more information on our requirements for LaTeX submissions can be found at http://journals.plos.org/plosone/s/latex.

Reviewers' comments:

Reviewer's Responses to Questions

**Comments to the Author**

1. Does the manuscript adhere to the experimental procedures and analyses described in the Registered Report Protocol?

If the manuscript reports any deviations from the planned experimental procedures and analyses, those must be reasonable and adequately justified.

Reviewer #1: Yes

Reviewer #2: Yes

2. If the manuscript reports exploratory analyses or experimental procedures not outlined in the original Registered Report Protocol, are these reasonable, justified and methodologically sound?

A Registered Report may include valid exploratory analyses not previously outlined in the Registered Report Protocol, as long as they are described as such.

Reviewer #1: Yes

Reviewer #2: Yes

3. Are the conclusions supported by the data and do they address the research question presented in the Registered Report Protocol?

The manuscript must describe a technically sound piece of scientific research with data that supports the conclusions. The conclusions must be drawn appropriately based on the research question(s) outlined in the Registered Report Protocol and on the data presented.

Reviewer #1: Partly

Reviewer #2: Yes

4. Have the authors made all data underlying the findings in their manuscript fully available?

Reviewer #1: Yes

Reviewer #2: Yes

5. Is the manuscript presented in an intelligible fashion and written in standard English?

Reviewer #1: Yes

Reviewer #2: Yes

6. Review Comments to the Author

Please use the space provided to explain your answers to the questions above. (Please upload your review as an attachment if it exceeds 20,000 characters)

Reviewer #1: ## Review PLOSONE: the impact of trait number and correlation on functional diversity metrics in real-world ecosystems

# General comments

I review the paper entitled "the impact of trait number and correlation on functional diversity metrics in real-world ecosystems" submitted to Plos ONE journal. In this paper, the authors deal with a timely question in functional ecology related to the choice of functional traits to calculate functional diversity indices. Indeed, given the development of trait measurements in ecology, trait-based approaches are more and more common to go behind taxonomic diversity. A key aspect of these approaches is the sensitivity of functional diversity indices to the choice of functional traits (e.g. number and correlation).

In this paper, the authors propose to test the sensitivity of 8 functional diversity indices on real community of plants. Thus, they test the behavior of functional diversity indices to the number of functional traits used to describe the community. In addition they also test how the traits correlation influence the FD indices. They found that some indices decrease when the number of traits increased, while other increase and some are not changing. They also found that the trait-trait correlation have no significant influence of the functional indices.

The more I read the paper the more I had some problems to understand the real aim of this paper. When the authors claim L. 75 "Therefore, functional diversity could differ among replicate plots or sites simply because of the number or types of traits used to calculate the metric without any underlying ecological basis." and L. 407: "Some metrics may be unreliable measures for comparing functional diversity among communities since comparisons are dependent upon the number of traits used to calculate them." OK, I agree, but does people really compared FD of communities calculated on different number of traits ? I would not do it in any case, whatever the metrics give similar or different values. It is just like comparing apple and pear.

As a general comment, I would say that the authors pointed out a real question in their introduction, but the analyses and interpretations are not sufficient to answer the question. The main problem here is the lack of analyses on the type of traits, only the number is taking into consideration. L.104 "The field lacks clear guidelines for researchers to follow when choosing the number and types of traits to include when calculating functional diversity metrics." Although this statement is not completely true, this study do not help to solve it. There is no conclusion on which traits and how many traits is required. So far, this study gives hint about sensitivity of indices on trait number. Moreover, there is no clear explanation about ecological processes behind, since analyses performed here pointed out mathematical properties of the different indices to trait number.

About the sensitivity of FD to trait number: This is a very exiting question ! However, I am not sure I found a satisfying answer in this paper, or at least given the analyses the authors performed here. The fact that FD indices varies with the number of traits is not surprising but more importantly, I do not know how to interpret it! The authors seem to interpret it as a weak point of the method since they focus on FDis (1st paragraph of discussion), which do not vary with the number of traits. They justify it by the fact that : "This suggests that FDis provides reliable values across different communities and sets of traits, making it a potentially valuable tool for assessing patterns of functional diversity across communities and ecosystems." This is not true. For me, what is important is the ranking of communities across trait number. The question of community ranking is the key aspect here. Indeed, more than the values itself, most of the study in community ecology are based of the relationship of FD indices between communities. I would like to see this part more developed.

Moreover, FD indices, at least some of them, are highly influenced by the species richness. Most of community ecologist working of FD used null models where they compared the observed values to expected values. This would be more interesting to analysis and interpret, because SES will be comparable between community irrespective of their species richness, but also between number of traits with clear null hypothesis that a useful SES values of a FD index should be stable.

From a methodological point of view, I am not sure whether the FD indices are calculated directly using traits or after PC(o)A. L. 280, the authors claim that they "used dimensionally reduction where necessary". but later in the paragraph, they said L.291 they "calculated each metric using all possible combination of two traits up to all possible combinations of the maximum number of traits". There is also no information on whether traits are center/scale before calculated FD indices. I got a bit lost here, or try to explain better what you want to do. This might have strong implications, since if not scaled, traits might have different weight.

If the authors used traits to calculate indices; I disagree with their discussion. L. 407, they said that some FD indices are not suitable because the ranking changes given the number of traits. and later, L.415: "This is particularly concerning given the often arbitrary nature of selecting the number of traits used in a study.". This is not surprising but instead, it can be useful, meaning that some traits (or combination) add some information. Otherwise why not using only body size... Moreover there is no explanation why such differences happened. Does some specific type of traits bring new information and change the patterns ? Here, and more generally in the discussion, there is a lack of ecological explanations of the results. A great advantage of working on real community would be to correlate outputs of indices (mathematics) to ecological process or at least to the different type of traits/species.

In addition, when we have several traits, a common strategy is to make a PCA/PCoA, then it would have been wise to test if the number of traits changes the results, not on the raw data but on the FD indices calculated after using a PCoA. FD indices would be calculated on a PCoA with same number of dimension. Such approach would be more relevant.

About the trait-trait analysis, the effect of traits correlation was calculated only with 4 traits (L.295), but I am a bit surprising of this choice. Please justify it. All this lack of details (including my previous remarks) make that the trait-trait analyses have weak support so far.

Finally, there is no mention of the trait selection. The choice of the traits and its selection seems to be randomized (e.g., L; 294: "10 different traits so there are 45 2-trait calculations, 120 3-trait calculations[...]). Do it mean that only the number but not the identity matter? And should we expect similar results if the type of traits is different ? For instance the authors L. 402 explain differences with other studies by the number of traits, but in any case we have information of the type of traits used.

# Minor comments

L. 46. Do you have example of such studies ? This is quite dangerous approach

L. 75. reference is needed

L. 75. "Therefore, functional diversity could differ among replicate plots or sites simply because of the number or types of traits used to calculate the metric without any underlying ecological basis." But if the trait differs, its means that the ecological processes also differ. here the problem is not about metrics, but more about what we mean by "functional diversity". In another way, it is NOT because we got similar result with different set of traits, that the metric is better (or true).

L. 81. "it is important to determine the conditions under which they reflect ecological processes as opposed to mathematical patterns." I agree, but this aspect is not solve in this paper. for that a analysis on the trait type and relevance with ecological processes is needed not on the metrics. Metrics is just a way to calculate but it will in any case represent ecological processes.

L. 80. "As use of multi-trait functional diversity increases, it is important to determine the conditions under which they reflect ecological processes as opposed to mathematical patterns." OK, but this is not what the authors are doing here.

L. 122. replace value by metrics

L. 137: To refine objective of the paper I suggest to clarify the objective 1 (L. 137) the term "vary" need to be better explained. Moreover, I do not understand well this objective since the expectations are mathematical properties of the indices. Maybe can be wise to adapt them to the specific trait/species of the grassland?

L. 228/ Are the traits scaled/centered?

L. 280/ What means "where necessary"?

L. 318: "Adequate" avoid such vague vocabulary.

L. 328: How many communities have only one species ?

L. 331: this sentence is not clear

L. 332: "some": which?

L. 336: "relationships of communities to each other" rephrase it

L. 332: The reference to null model is not clear here. Do you mean that the relationship is not significantly different that expected under null model?

L. 362: I did not understand the goal of this analysis here. it has not been mention in introduction, and barely in method.

L. 376. The authors focused on 1 index, but it is not explain why here.

L. 402. Here a discussion about the type of trait is also needed.

L. 407. I do not agree. Here some analyses of SES are required to conclude

L. 412. Can you provide some explanation why such discrepancies of FEve? Does it come from mathematical issues or ecological process?

L. 450: I do not agree about this distinction between continuous/categorical data. Continuous are more common and available on a larger range of species. Moreover filling gap procedure are also well-know and work quite well ! Instead I would expect a real discussion on why indices are working on both type of distance.

Fig: in some cases, not all communities are visible, maybe you can use thinner lines and/or transparency

Please check your references, some are in different style.

Sincerely,

Reviewer #2: I reviewed the manuscript ‘The impact of trait number and correlation on functional diversity metrics in real-world ecosystems’ submitted for publication in PLOS One. The authors use vegetation data from natural and experimental communities to explore how trait number and correlation impact on various functional diversity metrics. The Introduction is extremely well written and clearly explains the importance of this study. The Discussion is also great and provides useful guidelines for researchers analysing functional diversity. Since the paper is rather methodological I found that the Methods and Results part could benefit with some further explanations and clarifications to further enhance the clarity and readability of the study. Please see my suggestions below:

Methods:

Line 161 – it’s a little difficult to follow how many sites and grasslands there were (e.g. in line 163 could be understood that each grassland had a site with natural and a site with planted community), maybe start the site description with the fact that there were two sampling sites and then mention how many communities and whether natural or planted were sampled in each site

Lines 165, 173 – add approximate coordinates for the sites

Line 182 – add which year

Line 185 –unclear if the 48 plots were in total for four communities, or were there 48x4 plots? explain somewhere how the two communities were located and how they differed from each other

Line 205 – the text in line 202 gives impression that data was collected only in 2018, so it reads a little strange here that the 2018 was selected

Line 249 – remove one ‘then’ from the sentence

Line 258 – missing part of the formula

Line 266 – this is also mentioned in lines 276-277, maybe not needed here

Line 305 – unclear here how the models were run for each community if the min/max/mean trait correlations were also calculated for each community (so there’s a single value of explanatory variable)

Results

Lines 317-329 – this would fit better in the methods, some of this information is already mentioned earlier in the text

Line 332 – if i understand correctly then only null model shows no relationship? if that’s the case then use ‘i.e.’ instead of ‘e.g.’

Line 335 – are these results shown somewhere? add reference to figure

Line 336 – the part ‘relationships of communities to each other’ is unclear, maybe just use the phrasing that is currently in the parentheses (and maybe use ‘from low to high functional diversity metric values’ for clarity)

Line 344 – is this correct ‘depending upon the number of traits used to calculate them’? isn’t this the relationship that is being tested? maybe rephrase to clarify, consider also adding an explanation to the Methods what rankings were compared

Table 3

Unclear what is ‘functional form’.

Add some further explanations in the table header as to how to read the table, it’s not very intuitive that the models that are compared are per each metric, distance matrix and predictor combination (within a row, but considering the same distance matrix, right?), and not clear that the 6 (for each distance measure) models are for different communities.

Explain also what mean, maximum and minimum correlations mean, so the table would be understandable without necessarily reading the text.

Line 352 – add a reference to a table or figure

LIne 354 – use ‘an Euclidean’

Line 360 – unclear what is the percentage from, models?

Discussion:

Line 381 – the part ‘though their magnitude was affected by input trait number’ could be removed since this is said also at the end of the sentence and currently makes the sentence difficult to read

Line 412 – use ‘community CDR4’ to remind the reader again what these acronyms stand for

Line 428 – what is ‘and unresponsive relationship’, do you mean ‘or no relationships’

Line 444 – could this be better as ‘Gower or Euclidean dissimilarity’?

Fig. 1

Are some of the communities not shown, e.g. in J, or are they covered up by each other? Maybe with a bit thinner lines and smaller points they could be visible?

Supplementary materials

The data collection protocols include Konza, but since this data was not used in the current study, maybe not needed to include here to avoid confusion

7. PLOS authors have the option to publish the peer review history of their article (what does this mean?). If published, this will include your full peer review and any attached files.

Reviewer #1: No

Reviewer #2: No

---

## [Author Response · Author response to Decision Letter 0]

19 Apr 2024

We thank the two reviewers for their comments. Their reviews were most helpful for improving the clarity of the text and focusing the aims of our study. We have responded to each individual comment. In most cases, we agreed with the reviewers comments and either altered text or changed/added analyses as necessary. In a few instances for which additional analyses were proposed, we have provided justification for the methodology proposed in the Registered Report article and performed for this manuscript. For convenience, we have included in these responses quoted text from our most recent submission. Reviewers comments are shown in standard black text while our responses are in blue italicized text.

Reviewer #1: ## Review PLOSONE: the impact of trait number and correlation on functional diversity metrics in real-world ecosystems

# General comments

I review the paper entitled "the impact of trait number and correlation on functional diversity metrics in real-world ecosystems" submitted to Plos ONE journal. In this paper, the authors deal with a timely question in functional ecology related to the choice of functional traits to calculate functional diversity indices. Indeed, given the development of trait measurements in ecology, trait-based approaches are more and more common to go behind taxonomic diversity. A key aspect of these approaches is the sensitivity of functional diversity indices to the choice of functional traits (e.g. number and correlation).

In this paper, the authors propose to test the sensitivity of 8 functional diversity indices on real community of plants. Thus, they test the behavior of functional diversity indices to the number of functional traits used to describe the community. In addition they also test how the traits correlation influence the FD indices. They found that some indices decrease when the number of traits increased, while other increase and some are not changing. They also found that the trait-trait correlation have no significant influence of the functional indices.

The more I read the paper the more I had some problems to understand the real aim of this paper. When the authors claim L. 75 "Therefore, functional diversity could differ among replicate plots or sites simply because of the number or types of traits used to calculate the metric without any underlying ecological basis." and L. 407: "Some metrics may be unreliable measures for comparing functional diversity among communities since comparisons are dependent upon the number of traits used to calculate them." OK, I agree, but does people really compared FD of communities calculated on different number of traits ? I would not do it in any case, whatever the metrics give similar or different values. It is just like comparing apple and pear.

As a general comment, I would say that the authors pointed out a real question in their introduction, but the analyses and interpretations are not sufficient to answer the question. The main problem here is the lack of analyses on the type of traits, only the number is taking into consideration. L.104 "The field lacks clear guidelines for researchers to follow when choosing the number and types of traits to include when calculating functional diversity metrics." Although this statement is not completely true, this study do not help to solve it. There is no conclusion on which traits and how many traits is required. So far, this study gives hint about sensitivity of indices on trait number. Moreover, there is no clear explanation about ecological processes behind, since analyses performed here pointed out mathematical properties of the different indices to trait number.

Thank you for this comment as it has helped us clarify the aims of this study. Though we might have found a conclusive answer as to the specific number of traits required to generate a robust comparison of rankings among communities, our results generated no such obvious conclusion. We did report results and include recommendations for using certain metrics for which rankings among communities were inconsistent across the range of number of traits used to calculate them (such as FEve). We agree that the focus of this paper is primarily to test the mathematical properties of these metrics and we believe that our study represents meaningful progress in the field by, for the first time, using real grassland data in this investigation instead of simulated data.

About the sensitivity of FD to trait number: This is a very exiting question ! However, I am not sure I found a satisfying answer in this paper, or at least given the analyses the authors performed here. The fact that FD indices varies with the number of traits is not surprising but more importantly, I do not know how to interpret it! The authors seem to interpret it as a weak point of the method since they focus on FDis (1st paragraph of discussion), which do not vary with the number of traits. They justify it by the fact that : "This suggests that FDis provides reliable values across different communities and sets of traits, making it a potentially valuable tool for assessing patterns of functional diversity across communities and ecosystems." This is not true. For me, what is important is the ranking of communities across trait number. The question of community ranking is the key aspect here. Indeed, more than the values itself, most of the study in community ecology are based of the relationship of FD indices between communities. I would like to see this part more developed.

We agree that the relationships of FD indices among communities is the most important factor to consider. To that end, the first paragraph of the discussion section highlights the suite of metrics that maintain consistent relationships among communities across the range of trait numbers L 435-441: “Similarly, Rao's Q, KDE dispersion, and KDE richness maintained consistent ordered rankings of metrics among communities across the range of trait numbers for both Gower and Euclidean dissimilarity matrices, although differences between communities were magnified at higher numbers of traits when using Euclidean matrices. Other metrics (FEve, KDE evenness, FDiv) had less consistency across the range of traits used to calculate them as relative rankings of different communities changed with the number of traits used in constructing the metrics.“

L. 463-469: “Specifically, FRich, FEve, and KDE evenness showed crossing slopes among communities (i.e. ranking of communities changed with the number of traits) for both Gower and Euclidean distance matrices. The inconsistency of communities’ relationships to one another across the range of the number of traits raises concerns. For example, community CDR4 had a greater FEve than community SEV1 when using four traits, but CDR4 had a smaller FEve than SEV1 when calculated with eight traits. Such discrepancies have the potential to introduce discordant results in the literature, even when otherwise identical studies have been conducted.”

 In addition, our recommendations support the use of metrics that maintain consistent rankings of communities and suggest supplementary analyses for studies using metrics for which rankings among communities are inconsistent L 528-533: “Based on our findings, we recommend use of FDis, KDE dispersion, and/or Rao’s Q in analyses of functional diversity as all of these measures provide consistent results among communities at all numbers of traits tested. Additionally, due to the inconsistency of evenness metrics with respect to community rankings, we strongly recommend that any use of FEve or KDE evenness metrics include supplemental analyses to test whether results are consistent with different numbers of traits used to calculate them.“

Moreover, FD indices, at least some of them, are highly influenced by the species richness. Most of community ecologist working of FD used null models where they compared the observed values to expected values. This would be more interesting to analysis and interpret, because SES will be comparable between community irrespective of their species richness, but also between number of traits with clear null hypothesis that a useful SES values of a FD index should be stable.

We recognize that use of richness-based null models is helpful in some contexts in order to isolate the effects of traits on FD separate from effects of species richness. In this study, we specifically chose an approach agnostic to species richness as the focal objective was to resolve the predictability and reliability of metrics across ranges of trait number and correlations as opposed to investigating which metrics were more or less informative than simple species richness. Consider, if a species-based null model was chosen as the best model through model selection in our paper, we could conclude that the FD metric is a worse predictor than species richness but would have learned nothing about reliability of FD metrics. To that end, we do include a null, intercept-only models which test whether or not any change in metrics occurs across the variables which we test (number of traits and correlation). Many papers in the literature take a similar, trait-agnostic approach when questions focus on comparing functional diversity among communities as opposed to comparing relative explanatory power of richness among communities (e.g. Zihao et al. 2021, Mao et al. 2022). 

From a methodological point of view, I am not sure whether the FD indices are calculated directly using traits or after PC(o)A. L. 280, the authors claim that they "used dimensionally reduction where necessary". but later in the paragraph, they said L.291 they "calculated each metric using all possible combination of two traits up to all possible combinations of the maximum number of traits".There is also no information on whether traits are center/scale before calculated FD indices. I got a bit lost here, or try to explain better what you want to do. This might have strong implications, since if not scaled, traits might have different weight.

We have now added additional text to clarify our methodology. There seems to be confusion about dimensionality reduction and how we looked at the impact of trait number on the metrics. The text now reads: 

L 281-287: “To understand the impact of trait number on functional diversity, each functional diversity metric was calculated using all possible combinations of two traits up to all possible combinations of the maximum number of traits at each site. For example, at Sevilleta there are 10 different traits so there are 45 2-trait calculations, 120 3-trait calculations, 210 4-trait calculations, and so forth up to 10 9-trait calculations and 1 10-trait calculation. This allows us to focus on the impact of trait number independent of the constituent set of traits used to calculate the metric.”

L288-297: “To calculate the five metrics using the FD package, we first calculated a species-trait distance matrix using both Gower (categorical and continuous traits) and Euclidean (continuous traits only) distances. These distance matrices were calculated with both scaled and centered and non-scaled trait data for each community. Centering was done by subtracting the trait mean from each observation and scaling was done by dividing the centered traits by their standard deviations (as in the FD package). These distance matrices along with a species-abundance matrix are the input for the FD package. The FD package performs a principal components analysis on the full species-trait distance matrix. Dimensionality reduction only occurs for FRic and FDiv metric calculation. For all FRich and FDiv analyses, we hold the number of dimensions equal to 2, similar to Legras et al. [20].”

We have added a table to the supplemental material which shows the model selection results when unscaled data are used (i.e. Table 3 in the main text but using unscaled data). The best fit functional forms changed for just a few of the metrics and overall the changes do not affect our interpretation of the results.

If the authors used traits to calculate indices; I disagree with their discussion. L. 407, they said that some FD indices are not suitable because the ranking changes given the number of traits. and later, L.415: "This is particularly concerning given the often arbitrary nature of selecting the number of traits used in a study.". This is not surprising but instead, it can be useful, meaning that some traits (or combination) add some information. Otherwise why not using only body size... Moreover there is no explanation why such differences happened. Does some specific type of traits bring new information and change the patterns ? Here, and more generally in the discussion, there is a lack of ecological explanations of the results. A great advantage of working on real community would be to correlate outputs of indices (mathematics) to ecological process or at least to the different type of traits/species.

Thank you for this comment. We agree that the primary results from this work represents mathematical properties of these indices and therefore we do not try to make unsupported ecological conclusions. In our analyses, every combination of traits exists at each of the values of number of traits on the x-axis. Therefore, since every trait is equally represented across the range of x-axis values, the fact that the values of metrics change across the range of values is not due to ecological processes of different traits being used to calculate the metrics. In other words, the trait-number analyses do not assess relative contribution of certain traits because it is agnostic of trait identity. Conveniently, the trait correlation analyses provide information as to how the types of traits used impact the metrics, albeit with a focus on trait correlation as opposed to trait identity per se. Though we can speculate as to how certain traits drive variability in certain metrics (e.g. line 446-450 which reads “These differences in findings might be attributed to the inherent complexity and noise present in real-world data. Real communities often contain anomalous species with outlier trait values (e.g. a gymnosperm among angiosperms, a tree seedling among herbaceous plants, or other rare species outlier values), which can exert considerable influence on evenness indices.“), conclusions about how these metrics relate to ecological properties or specific conclusions about certain traits or species are outside of the scope of this work and would require a wholly different study design. However, we agree on the importance of such work.

In addition, when we have several traits, a common strategy is to make a PCA/PCoA, then it would have been wise to test if the number of traits changes the results, not on the raw data but on the FD indices calculated after using a PCoA. FD indices would be calculated on a PCoA with same number of dimension. Such approach would be more relevant.

Though use of PCA/PCoA for dimensionality reduction is common in studies using functional diversity metrics, it is also common to not use dimensionality reduction (e.g. Thakur & Chawla 2019, Zuo et al. 2021, Biswas et al. 2019, Niu et al. 2015). In particular, KDE metrics do not use dimensionality reduction in their calculation and in order to make metrics most comparable to each other, we also did not use dimensionality reduction in the FD metrics except for those metrics that require it. Moreover, Legras et al. 2020 showed in their supplemental material that metrics are very similar regardless of the extent of dimensionality reduction.

About the trait-trait analysis, the effect of traits correlation was calculated only with 4 traits (L.295), but I am a bit surprising of this choice. Please justify it. All this lack of details (including my previous remarks) make that the trait-trait analyses have weak support so far.

Four traits was chosen as a compromise between the minimum number of traits likely to be used to calculate these indices and an effort to reduce noise in the evaluation of minimum and maximum trait correlation. Max and min include all possible combinations of four traits and therefore all possible pairwise combinations of two traits to generate max and min. Including greater numbers of traits in these a

---

## [Decision Letter · Decision Letter 1]

23 May 2024

PONE-D-24-03515R1The impact of trait number and correlation on functional diversity metrics in real-world ecosystemsPLOS ONE

Dear Dr. Ohlert,

Thank you for submitting your manuscript to PLOS ONE. After careful consideration, we feel that it has merit but does not fully meet PLOS ONE’s publication criteria as it currently stands. Therefore, we invite you to submit a revised version of the manuscript that addresses the points raised during the review process.

We look forward to receiving your revised manuscript.

Kind regards,

Francesco Boscutti

Academic Editor

PLOS ONE

Journal Requirements:

Additional Editor Comments:

Reviewers' comments:

Reviewer #2:

 The authors have done a great job in revising the paper and addressing reviewers' comments. I only have a few minor suggestions to improve the clarity of the text in a few places:

Line 50 – the acronyms are not defined in the abstract, better to write out or define in lines 44-45

Table 3

Line 375 – in the table header the example for Frich, Trait number and Gower (first line in the table) doesn’t fit with the results in the table (text says 2 for linear and 4 for quadratic, but table has 3 and 3 – i think the text hasn’t been updated for the new version of the table)

In the table use ‘FRich’, ‘FDis’, ‘FEve’ and ‘FDiv’ like in the rest of the manuscript

KDE diversity measures are not capitalised in the the text (e.g. ‘evenness’), but are here and on the figures – better to use the same spelling everywhere

Line 385 – this is the first time CDR is used, but it’s not defined anywhere

Fig. 1 and Fig. 2 have different legends, but I guess these should be the same?

Fig. 2, Fig S1, Fig S2

Lines 399, 407, 415 – CDR and SEV are not defined

Line 527 – perhaps this section could be combined with the previous one (e.g. using heading ‘The ecological significance of methodology and recommendations’ or something similar) since some recommendations are already given in lines 519-526

---

## [Author Response · Author response to Decision Letter 1]

3 Jun 2024

Reviewers' comments:

Reviewer #2:

 The authors have done a great job in revising the paper and addressing reviewers' comments. I only have a few minor suggestions to improve the clarity of the text in a few places:

Line 50 – the acronyms are not defined in the abstract, better to write out or define in lines 44-45

We agree and have made this change.

“We found that most metrics were sensitive to the number of traits used to calculate them, but functional dispersion (FDis), kernel density estimation dispersion (KDE dispersion), and Rao’s quadratic entropy (Rao’s Q) maintained consistent rankings of communities across the range of trait numbers” lines 43-46

Table 3

Line 375 – in the table header the example for Frich, Trait number and Gower (first line in the table) doesn’t fit with the results in the table (text says 2 for linear and 4 for quadratic, but table has 3 and 3 – i think the text hasn’t been updated for the new version of the table)

In the table use ‘FRich’, ‘FDis’, ‘FEve’ and ‘FDiv’ like in the rest of the manuscript

KDE diversity measures are not capitalised in the the text (e.g. ‘evenness’), but are here and on the figures – better to use the same spelling everywhere

We agree and have made these consistent.

Line 385 – this is the first time CDR is used, but it’s not defined anywhere

We have now defined CDR and SEV in lines 170 and 178, respectively.

Fig. 1 and Fig. 2 have different legends, but I guess these should be the same?

These are now consistent.

Fig. 2, Fig S1, Fig S2

Lines 399, 407, 415 – CDR and SEV are not defined

We have now defined these in lines 170 and 178.

Line 527 – perhaps this section could be combined with the previous one (e.g. using heading ‘The ecological significance of methodology and recommendations’ or something similar) since some recommendations are already given in lines 519-526

We like this idea and have edited it accordingly. Line 515

---

## [Editor Report · Decision Letter 2]

11 Jun 2024

The impact of trait number and correlation on functional diversity metrics in real-world ecosystems

PONE-D-24-03515R2

Dear Dr. Ohlert,

We’re pleased to inform you that your manuscript has been judged scientifically suitable for publication and will be formally accepted for publication once it meets all outstanding technical requirements.

Kind regards,

Francesco Boscutti

Academic Editor

PLOS ONE
---

## [Editor Report · Acceptance letter]

20 Jun 2024

PONE-D-24-03515R2 

PLOS ONE

Dear Dr. Ohlert, 

I'm pleased to inform you that your manuscript has been deemed suitable for publication in PLOS ONE. Congratulations! Your manuscript is now being handed over to our production team.

Kind regards, 

on behalf of

Dr. Francesco Boscutti 

Academic Editor

PLOS ONE